# Development of a core outcome set for behavioural weight management programmes for adults with overweight and obesity: protocol for obtaining expert consensus using Delphi methodology

Ruth M Mackenzie,[1] Louisa J Ells,[2] Sharon Anne Simpson,[3] Jennifer Logue [1]

[1]Institute of Cardiovascular and Medical Sciences, University of Glasgow, Glasgow, UK
[2]School of Health and Social Care, Teeside University, Middlesbrough, UK
[3]Institute of Health and Wellbeing, University of Glasgow, Glasgow, UK

**Correspondence to**
Dr Jennifer Logue;
Jennifer.Logue@glasgow.ac.uk

## ABSTRACT

**Introduction** Weight management interventions in research studies and in clinical practice differ in length, advice, frequency of meetings, staff and cost. Very few real-world programmes have published patient-related outcomes, and those that have published used different ways of reporting the information, making it impossible to compare interventions and further develop the evidence base. Developing a core outcome set for behavioural weight management programmes (BWMPs) for adults with overweight and obesity will allow different BWMPs to be compared and reveal which interventions work best for which members of the population.

**Methods and analysis** An expert group, comprised of 40 people who work in, refer to, or attend BWMPs for adults with overweight and obesity, will be asked to decide which outcomes services should report. An online Delphi process will be employed to help the group reach consensus as to which outcomes should be measured and reported, and which definitions/instruments should be used in order to do so. The first stage of the Delphi process (three rounds of questionnaires) will focus on outcomes while the second stage (three additional rounds of questionnaires) will focus on definition/instrument selection.

**Ethics and dissemination** Ethical approval for this study has been received from the University of Glasgow College of Medical, Veterinary and Life Sciences Ethics Committee. With regard to disseminating results, a report will be submitted to our funding body, the Chief Scientist Office of the Scottish Government Health Department. In addition, early findings will be shared with Public Health England and Health Scotland, and results communicated via conference presentations, peer review publication and our institutions' social media platforms.

## Strengths and limitations of this study

► The major strength of this study is that it is the first of its kind and development of a core outcome set for behavioural weight management programmes for adults with overweight and obesity is much needed in order to standardise reporting which, in turn, will lead to a better evidence base and improvements in weight management provision.

► It is a limitation that this study is wholly based in the UK as the results may need some adaptation to be suited to real-world programmes set within other healthcare systems.

► The recognised method for core outcome set development, the Delphi method, will be used to garner opinions from a wide range of individuals with expertise in behavioural weight management.

► Review of all existing qualitative research studies will not be undertaken when generating the initial list of outcomes. However, qualitative work will be performed during core outcome set development as part of the Delphi process.

interventions. These have proven efficacy from randomised controlled trials.[3] However, their implementation in practice is inconsistent with mapping exercises in Scotland[4] and England[5] showing wide variation in services in terms of inclusion criteria, referral routes, delivery format, length and cost. Few real life services have published data and when they do publish, results can be poor with low levels of completion and 'success', and lack of longer term outcomes.

The NICE guidance, 'Weight management: lifestyle services for overweight or obese adults',[1] identified a number of evidence gaps. These included, reliance on studies with short follow-up, collection of data at limited time points, small sample sizes, demographic samples that limit the ability to generalise, non-reporting of reasons for people

## INTRODUCTION

Both the National Institute for Health and Care Excellence (NICE)[1] and Scottish Intercollegiate Guidelines Network[2] guidelines outline the intervention components to be included in a community weight management programme, namely calorie restriction, increased physical activity and behavioural

dropping out and lack of evidence regarding the effect of population characteristics, such as age, gender and socio economic status, on the effectiveness of a service. They noted a lack of comparisons between behavioural weight management programmes (BWMPs) in the UK. This lack of an evidence base means that it is not possible to issue clear guidance as to which services are cost effective for which population groups.

Public Health England (PHE) has created a standard evaluation framework (SEF)[6] to aid the evaluation of real-world weight management programmes. However, in their 2015 weight management mapping exercise,[5] PHE reported that only 46% of adult weight management programmes use the SEF and, as it simply suggests areas for reporting and potential methods of analysis, there is a huge gap in standardised reporting. PHE had intended to analyse data from services but analysis was not possible due to the heterogeneity of reporting which included kilograms, % weight loss, average number of completers achieving 5% wt loss, body mass index (BMI) and more.[5] With regard to research studies, evidence suggests similar heterogeneity in terms of the reporting of outcomes.[7]

In an attempt to address this reporting issue, PHE issued a minimum dataset[8] which provides an important core outcome recommendation for England, stipulating collection of certain demographics, service details, BMI and well-being at baseline, on completion of the programme and at 6 months and 12 months postprogramme. A data collection tool provides information to support the standardisation of these data collection practices. This minimum dataset will be used to support PHE's recently released document on adult tier 2 wt management service key performance indicators (KPIs)[9] which provides advice as to how weight status and service compliance should be reported and measured.

The study described herein has been funded through a Chief Scientist Office of the Scottish Government Health Department grant and will serve to further validate and build on the PHE minimum dataset[8] and KPI document,[9] while also informing a similar framework for Scotland. In addition, our research will provide much needed consensus on the measurements that should be used, such as questionnaires, something currently not covered in the PHE minimum dataset[8] or KPI document.[9] Overall, this work will ensure more consistency in the measurement of the effectiveness of adult weight management services, leading to a better evidence base from which to identify which services are effective across a range of settings.

Recently, a core outcome set for bariatric and metabolic surgery was successfully developed using consensus methodology.[10] However, outcomes, including perioperative outcomes and postoperative complications, are not relevant for reporting from BWMPs. Therefore, the aim of this study, which will run from November 2017 until November 2018, is to gain expert consensus opinion on the core outcomes that should be reported from behavioural weight management interventions for adults

with overweight and obesity in real-world clinical practice as well as within research studies.

The specific study objectives are to:
1. Review the list of outcomes previously reported in the PHE SEF,[6] minimum dataset[8] and KPI document[9];
2. Identify additional outcomes reported in studies of structured, sustained, multicomponent weight management programmes for adults from a systematic review of the literature;
3. Select outcomes for inclusion in the core dataset using consensus methodology;
4. Select definitions/instruments for measuring chosen outcomes using consensus methodology.

## METHODS AND ANALYSIS
### Identification of outcomes

We will generate a list of outcomes by review of the PHE SEF,[6] which was itself developed from a systematic review of the literature/focus groups, and from the PHE minimum dataset[8] and KPI document[9] which were developed through expert consensus and evidence from the peer review and grey literature.

Further outcomes will be selected by a review of included studies in the systematic review, 'The clinical effectiveness of long-term weight management schemes for adults' by Hartmann-Boyce et al,[7] conducted during the development of NICE guidance.[1] This systematic review[7] assessed the effects of multicomponent BWMPs in overweight and adults with obesity which may be applicable in the UK. To be considered a multicomponent BWMP, the components of the programme had to include diet, physical activity and behavioural therapy (for example, counselling sessions). The scope included commercial weight loss programmes and non-commercial programmes, such as those delivered in primary care settings (for example, in General Practice).[7] It updated and expanded on an existing systematic review published in 2011 by Loveman et al[3] and used similar methods. The Loveman systematic review[3] sought to assess the long-term clinical effectiveness and cost-effectiveness of multicomponent weight management schemes for adults in terms of weight loss and maintenance of weight loss.

Additional outcomes will be identified by updating the Hartmann-Boyce systematic review,[7] using the same inclusion criteria but extending search dates so that studies from 1/11/2012 until 30/09/17 are included. Search and selection criteria for the systematic review are identical to those of Hartmann-Boyce.[7] With regard to database searches, Hartmann-Boyce[7] searched BioSciences Information Service of Biological Abstracts (BIOSIS), the Cochrane Database of Systematic Reviews, Cochrane Controlled Register of Trials (CENTRAL), the Conference Proceedings Citation Index, the Database of Abstracts of Reviews and Effects, Embase, the Health Technology Assessment database, Medline, PsychInfo and Science Citation Index for references relating to weight loss programmes. They also screened references

from three additional sources: reference lists in systematic reviews, documents received via the NICE call for evidence and studies excluded from Loveman[3] that they wished to re-examine. Studies selected for inclusion had to be structured, sustained, multicomponent adult weight management programmes with interventions which were a combination of diet and physical activity with a behaviour change strategy to influence lifestyle. In addition, programmes were required to include a follow-up of more than 12 months and be delivered in the health sector, in the community or commercially (ie, applicable to the National Health Service; NHS).

Two review authors will independently assess the abstracts of studies resulting from our literature search. Full text copies of studies appearing to meet the inclusion criteria will be further independently assessed by the two reviewers. Following discussion, agreement will be reached as to which studies to include. Any new outcomes will then be identified from the selected studies from both Hartmann-Boyce[7] and the updated review.

### Identification of instruments
By review of the studies identified during the systematic reviews previously described, we will list instruments and definitions for selected outcomes. The study investigators will review this list and add any further suitable instruments.

### Data analysis and presentation
For analysis purposes, the data will be tabulated so that the outcomes and instruments to be included in our Delphi are listed and the study/studies from which they were identified are displayed. Outcomes and instruments will be grouped under appropriate domains following review of selected outcomes.

### Patient and public involvement
We will develop our core outcome set by means of consensus from an expert group. The sampling frame will aim to include members of the public with experience of NHS, local authority or commercial adult BWMPs in the UK, academics/policy makers/commissioners working in weight management, staff currently involved in delivering a BWMP for adults (without significant policy involvement) and primary care staff (referrers). Consensus methodology will ensure that the opinions and preferences of members of the public will be given the same weighting as those of the other experts.

There is no published agreement on the optimal size of an expert group; pragmatism is required while ensuring a range of opinions is garnered. Experience suggests a greater than 80% completion rate of Delphi questionnaires.[10 11] We will preapproach potential volunteers to get agreement to participate from 10 members of the public, 20 academics/policy makers/commissioners, 20 wt management staff and 10 primary care staff. Forty experts will complete each of the two separate Delphi processes.

For the first Delphi process (stage 1, outcome selection), 10 members of the public, 10 academics/policy makers/commissioners, 10 wt management staff and 10 primary care staff will be invited to participate.

For the second Delphi (stage 2, instrument selection), 20 academics/policy makers/commissioners and 20 wt management staff will be invited to participate with further members recruited if any of the original group (the 10 from each group who completed stage 1) have dropped out after the stage 1 Delphi. The stage 2 Delphi will involve reading papers, looking at metrics and assessing validity of instruments/questionnaires. As in depth knowledge of academic literature and reporting tools is required, this stage of the Delphi process will be restricted to academics/policy makers/commissioners and weight management staff.

A small monetary incentive (a £35 gift voucher for either John Lewis or Amazon, depending on preference) will be offered to members of the public and primary care staff as this study is not of any direct benefit to them and could not be considered part of their role.

Staff working in weight management, academics/policy makers/commissioners and primary care staff will be recruited by email from the investigators and their personal contacts, and also via an email from the Association for the Study of Obesity. An information letter outlining the study will be attached to emails. On registering interest in our study, we will ask volunteers from these groups to provide us with information as to their role and geographical location within the UK.

Members of the public will be recruited by email from the Association for the Study of Obesity (which has lay members) and from professional contacts (a number of weight management programmes have lay members on steering committees). An information letter outlining the study will be attached to emails. (The information letter for the public will be written in lay language and will therefore differ slightly to the information letter for the other groups.) We have also registered with the NIHR People in Research website (https://www.peopleinresearch.org/) where our study will be advertised (following review to ensure suitability for a lay audience). Our information letter will be available to download from this website. On registering interest in our study, a 'job description' pro forma will be sent to members of the public via email. They will be asked to complete this pro forma and return it to us by email. The pro forma will provide us with information as to their gender, age, geographical location and experience of BWMPs.

In addition, Facebook and Twitter will be used to recruit members of the public, weight management staff, academics/policy makers/commissioners and primary care staff. Facebook posts and Tweets will link to a Mailchimp recruitment page where volunteers will be able to register their interest. On doing so, they will receive the appropriate information letter. Weight management staff, academics/policy makers/commissioners and primary care staff will be asked to provide us

with information as to their role and geographical location within the UK, and members of the public will be asked to complete the job description pro forma.

Following provision of information regarding role and geographical location from weight management staff, academics/policy makers/commissioners and primary care staff, and the return of completed pro formas from members of the public, selection of volunteers to participate will commence. Selection will be based on our sampling framework which is outlined below. Volunteers will be sent an email to thank them for their interest and inform them if they have been selected to participate or not. A list of selected volunteers' names and email addresses will then be sent to Clinvivo (www.clinvivo.com, a spin-out company of the University of Warwick) who will be conducting the Delphi process. Clinvivo will then contact these individuals by email, providing a link to the online Delphi questionnaire and instructions as to how to complete it.

On completion of the study, all participants (including members of the public) will be sent (by email) a copy of the final outcome and definition/instrument sets. In addition, where consent has been given, participants (including members of the public) will be named as contributors in the results publication.

### Sampling framework

To ensure our volunteers are a representative UK group, of the 20 wt management staff selected, at least 50% will be from England. Similarly, at least 50% of the 20 academic/policy maker/commissioner group will be from England. Eight of the 20 (40%) will be academics, six of the 20 (30%) will be policy makers and six of the 20 (30%) will be commissioners. At least 50% of the 10 primary care staff selected will also be from England. With regard to

members of the public, more than 50% will have experience of commercial BWMPs, more than 50% will be of working age, more than 30% will be male and less than 30% will be from any one region of the UK.

### Delphi survey

In order to develop our core outcome dataset, Delphi methodology will be used to gain consensus from our expert group. Two Delphis (stage 1 and stage 2) will be carried out using an online system developed and conducted by Clinvivo. Each Delphi will be carried out online over three sequential rounds with the same group of participants (figure 1). For both stage 1 and stage 2 Delphis, only those who complete a questionnaire in round 1 will be eligible to participate in round 2, and only those who complete round 2 will be eligible to participate in round 3.

The stage 1 Delphi will involve asking each expert to score the importance of an outcome measure for use in weight management service outcome reporting. The scale will run from 1 to 9 with 1–3 indicating that the outcome is unimportant, 4–6 indicating that it is neither unimportant nor important and 7–9 indicating that it is important.

During the stage 2 Delphi, experts will be asked to score the appropriateness of outcome definitions and instruments for measurement of outcomes. Again, this will be done using a 1–9 scale with 1–3 indicating that the definition/instrument is inappropriate, 4–6 indicating that it is neither appropriate nor inappropriate and 7–9 indicating that it is appropriate.

### Statistical analysis

To assess disagreement and importance/appropriateness (and thus define consensus) the Research and

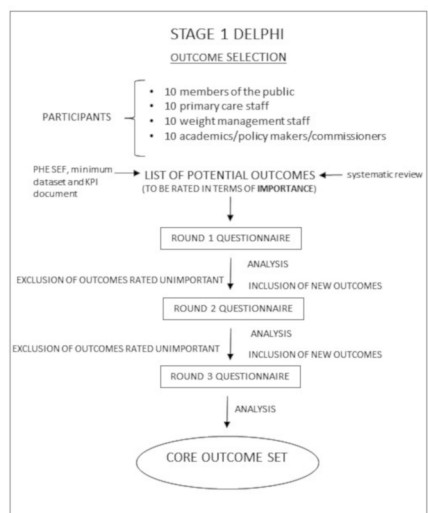
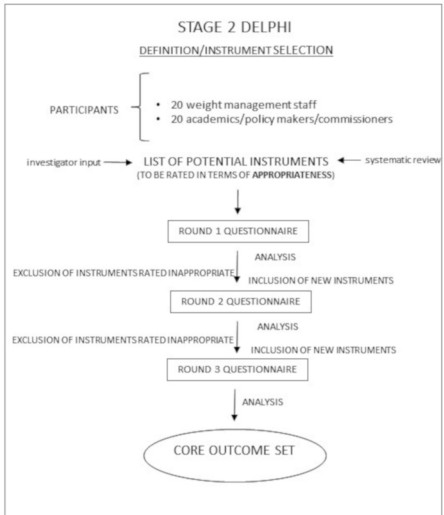

**Figure 1** Schematic outlining the two stage Delphi study. In order to develop a core outcome set and definition/instrument set, Delphi methodology will be used to gain consensus from expert groups. Two Delphis (stage 1 and stage 2) will be carried out online over three rounds of questionnaires. The stage 1 Delphi will focus on development of a core outcome set. The stage 2 Delphi will focus on corresponding definition/instrument selection. KPI, key performance indicator; PHE, Public Health England; SEF, standard evaluation framework.

Development (RAND)/University of California Los Angeles (UCLA) appropriateness method will be used.[11] This involves calculating the median score, the interpercentile range (IPR, 30th and 70th) and the interpercentile range adjusted for symmetry (IPRAS), for each item being rated.

Fitch *et al*[11] first explored using the IPR alone in an attempt to develop a method that reproduced 'classic' RAND definitions on panels that were multiples of three (which was typical in RAND's early consensus studies), but could also be extended to larger panel sizes. They found that in cases when agreement was good, the IPR should be narrow and in cases where there was disagreement, the IPR should be wide. However, an in-depth examination of the cases of disagreement identified by the IPR led to the discovery that when the ratings were symmetric, the IPR required to label an indication as disagreement was smaller than when the ratings were asymmetric, with respect to the middle. To overcome this, they developed the IPRAS which includes a correction factor for asymmetry (e*quation 1*).

### Equation 1

IPRAS=IPRr + (AI x CFA)

Where IPRr is the IPR required for disagreement when perfect symmetry exists, AI is the asymmetry index and CFA is the correction factor for asymmetry.

The IPRAS is the threshold beyond which the IPR for a particular item indicates disagreement. Using the IPRAS and the IPR to judge disagreement reproduces 'classic' RAND definitions when applied to panels made up of multiples of three, but can also be applied to panels of any size.[11] Variations on the stringencies of definitions of disagreement exist[12] but similar examples of Delphi studies in health services research have used the classic definition.[13–18] In *equation 1,* the optimal values for IPRr and CFA were derived following empirical work on a 9-point scale.[11] Fitch *et al* found that using values of 2.35 and 1.5 best reproduced the 'classic' definitions of agreement. These values will be used in this analysis. We will calculate AI as the distance between the central point of the IPR (p30 +p70/2) and the central point of the scale (ie, 5 on a 1–9 point scale.).

The IPRAS threshold is dependent on the symmetry of ratings about the median. Thus, each item requires a different IPRAS to be calculated. Consequently, the $i^{th}$ indication is rated with disagreement if the $IPR_i > IPRAS_i$. In previous Delphi studies some have calculated the ratio of these: the disagreement index.[14 16 18] If the disagreement index was less than 1.0, it indicated there was no disagreement for the item in question. However, this is problematic in terms of interpretation because in the case that the IPR is zero, then the ratio is zero, which can cause confusion. For this reason we will present IPR and IPRAS values and simply comment on whether or not there is disagreement (ie, when $IPR_i > IPRAS_i$).

Judgement of appropriateness/importance also follows the classic RAND definitions, and this is assessed simply as whether the median rating falls between 1 to 3 (inappropriate/unimportant), 4 and 6 (unsure) or 7 and 9 (appropriate/important).

At the end of each Delphi round, the median rating will be determined for individual outcomes/instruments and the distribution of ratings summarised in analysis conducted by Clinvivo and transferred to our research group (figure 1).

During both stage 1 and stage 2, participants will be given 2 weeks to complete each round of the Delphi and will be reminded of the deadline for completion before starting the process. Participants will also be sent a reminder email 1 day before the deadline for each round.

### Stage 1, round 1 Delphi

The first Delphi study (stage 1) will be to select outcomes for inclusion in the core dataset. Full instructions will be provided to the expert group prior to completion of stage 1 questionnaires. Outcomes will be grouped under appropriate domains (broadly based on the PHE SEF[6] and broadly following the weight management chronological pathway) and full definitions of each domain and outcome will be provided in lay language. Participants will be asked to rate each outcome in turn using the 1–9 scale. During round 1, there will be an option for adding free text outlining reasons for any given rating and also for suggesting possible additional outcomes.

### Analysis of stage 1, round 1

Additional outcomes listed by participants will be reviewed by two members of the study team (RMM and JL) to ensure they represent new outcomes. All outcomes, excluding any rated unimportant by consensus and including any new outcomes, will be carried forward to round 2.

### Stage 1, round 2 Delphi

In round 2, all experts will be asked to rate outcomes again. They will be shown their previous rating, the median expert group rating and any free text comments in the hope of ratings reaching a consensus. Experts will be asked to strongly consider the priority outcomes for weight management reporting in this round. Additional questions will be added as to the appropriate number of items to be included in the core outcome set.

### Analysis of stage 1, round 2

All outcomes, excluding any rated unimportant by consensus and including any new outcomes, will be carried forward to round 3.

### Stage 1, round 3 Delphi

In round 3, all experts will be asked to rate outcomes for the final time. They will be shown their previous rating, the median expert group rating and any free text comments in the hope of ratings reaching a consensus. Should it be the case that a large number of outcomes are being rated as important at this stage, the need to decide which outcomes should take priority for weight management

reporting will be reinforced to experts and they will be asked to rate only these priority outcomes as important. This will ensure development of a core outcome set of a manageable/practical size.

### Analysis of stage 1, round 3

Using the consensus on the outcome set size and importance of outcomes, an outcome set will be developed by the study team using the results of the Delphi.

### Stage 2, round 1 Delphi

The second Delphi study (stage 2) will be for definition/instrument selection. Selection of instruments for inclusion in the stage 2 Delphi will be informed, as previously stated, by results/ratings/suggestions from stage 1, systematic review and input from co-investigators (LJE and SAS).

Full instructions will be provided prior to completion of stage 2 questionnaires. As per stage 1, instruments will be grouped under appropriate domains and full definitions of each instrument will be provided. As stated, participants will be asked to rate each instrument in turn using a 1–9 scale of appropriateness (rather than importance). During the first round of the stage 2 instrument selection process, there will be an option for adding text outlining reasons for any given rating and also for suggesting possible additional instruments for measuring or defining outcomes.

### Analysis of stage 2, round 1

Additional instruments listed by participants will be reviewed by two members of the study team (RMM and JL) to ensure they represent new instruments. All instruments, excluding those rated inappropriate by consensus and including any new instruments, will be carried forward to round 2.

### Stage 2, round 2 Delphi

In round 2, all experts will be asked to rate instruments again. They will be shown their previous rating, the median expert group rating and any free text comments in the hope of ratings reaching a consensus. Experts will be encouraged to rate instruments in a way that shows their preferences.

### Analysis of stage 2, round 2

It may be that after round 2 an instrument set can be formed. Only those instruments related to an outcome for which there is no established consensus will be carried over to round 3.

### Stage 2, round 3 Delphi

In round 3, all experts will be asked to select instruments for the final time. They will be shown their previous rating, the median expert group rating and any free text comments in the hope of ratings reaching a consensus. In this round they will be asked to select the most appropriate instrument for each outcome in a binary format.

### Analysis of stage 2, round 3

A final instrument set matched to the core outcome set will be formed based on the consensus. In any areas where there is no consensus, the study team will adjudicate, taking account of free text comments.

### Data storage

Participants' contact details, including email addresses and telephone numbers, and the answers they provide, will only be stored by Clinvivo for the duration of the study. Clinvivo will not share participants' contact details with any third parties and participants' answers will be stored anonymously. Data will be encrypted before being stored on Clinvivo's server and prior to being transferred to the University of Glasgow. On completion of the study, Clinvivo will destroy all data after transferring it to the University of Glasgow. The University will securely store the data on password access computers for a period of 10 years following completion of the research project.

### Dissemination

With regard to disseminating the results of our study, we will communicate our results via peer review publication, conference presentations, professional societies and also via our institution's social media platforms.

In addition, we will submit a report to our funding body, the Chief Scientist Office of the Scottish Government Health Department. We will also share early findings with PHE and Health Scotland. We will be in full discussion with both bodies to ensure that our work informs their evaluation plans for BWMPs for adults with overweight and obesity.

Our study is, of course, restricted to the UK. This is due to BWMPs and their settings within health services being fairly country specific. For example, in France and the Netherlands there is no health insurance funding of BWMPs and, in the USA, obesity services are tertiary, combining behavioural programmes with medication and bariatric surgery. In addition, instruments, such as language and health economic models, can be country specific. Therefore, if used in an international context for trials or real-world services, our core outcome and definition/instrument set may require further adaptation.

**Contributors** RMM and JL drafted the protocol. LJE and SAS critically reviewed the protocol. RMM and JL finalised the protocol.

**Funding** This work was supported by the Chief Scientist Office of the Scottish Government Health Department, grant reference number CGA/17/08. SAS was supported by a MRC Strategic Award (MC-PC-13027, MC_UU_12017_14 and SPHSU14).

**Competing interests** JL leads a joint working project between University of Glasgow, NHS Greater Glasgow and Clyde, MSD and Astra Zeneca. The project also involved an educational grant from Janssen. JL received funding to attend a conference from Novo Nordisk.

**Patient consent for publication** Not required.

**Ethics approval** Ethical approval for this study has been received from the University of Glasgow College of Medical, Veterinary and Life Sciences Ethics Committee.

**Provenance and peer review** Not commissioned; externally peer reviewed.

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
