## [Reviewer comments · BMJ Open]

This paper was submitted to a another journal from BMJ but declined for publication following peer review. The authors addressed the reviewers' comments and submitted the revised paper to BMJ Open. The paper was subsequently accepted for publication at BMJ Open.

(This paper received three reviews from its previous journal but only two reviewers agreed to published their review.)

ARTICLE DETAILS

TITLE (PROVISIONAL)	Development of a core outcome set for behavioural weight management programmes for adults with overweight and obesity: protocol for obtaining expert consensus using Delphi methodology
AUTHORS	Mackenzie, Ruth M.; Ells, Louisa; Simpson, Sharon Anne; Logue, Jennifer

VERSION 1 – REVIEW

REVIEWER	Jamie Hartmann-Boyce University of Oxford, UK
REVIEW RETURNED	23-Jul-2018

GENERAL COMMENTS	This study is much needed and will no doubt make a valuable contribution to this field of research - I commend the authors on carrying this out. My comments are attached under separate cover (note - I am not familiar with the statistical methods for evaluating Delphi surveys and therefore am unable to comment on these, though from a non-expert point of view they seem appropriate). Most of my comments are minor and are "nice-to-have" rather than "need-to-have"; however, I do think increasing the level of involvement from patients and members of the public would markedly strengthen this work, and so have suggested this as a major revision, as I am keen that the authors consider this (and if resources don't allow further involvement, explicitly state this). Overall, though, I think this study will make a valuable contribution regardless of whether my suggested changes are made. Study Protocol for Developing a Core Outcome Set for Lifestyle Weight Management Programmes by Expert Consensus – Peer review comments General (throughout) There appears to be growing consensus that it is more appropriate to refer to these types of programmes as “behavioural weight management programmes” as opposed to “lifestyle weight management programmes”, as the latter is slightly more vague and difficult to interpret. Would suggest the authors consider changing this terminology. I think the findings from this exercise will be more valuable if patients/members of the public are included in a more robust way. I have the following suggestions: 1. include public advisors as part of project team, particularly to review recruitment materials and Delphi survey text before it is
--

	sent out to ensure it is written in accessible language (experience from other studies suggests when academics write in “lay language” it can still be a bit tricky to interpret!); 2. search to see if any work has been done with patient groups to establish outcomes or key questions in this area, rather than just relying on published literature and work from public health bodies (e.g. James Lind Alliance – if such work does not exist and resource permits the authors may want to consider a focus group with members of the public to identify new outcomes that aren’t in the literature, as I think doing this via Delphi may be challenging); 3. involve patients (and primary care staff) in second stage of Delphi (their opinions on measurement seem valid and I’m not clear why you haven’t included these groups); 4. Consider involving 20 (rather than 10) members of public in the Delphi exercise. Specific Abstract: “very few real-world programmes have published outcomes for patients” – suggest rewording – do you mean very few have published outcomes for patient consumption (e.g. an outcome report aimed at patients) or that very few have published patient-related outcomes? Strengths and limitations of this study – suggest removing “minor” from the phrase “some minor adaptation” as without conducting a broader exercise it is difficult to anticipate the extent of adaptation that may be needed Stage 1 Delphi process – further explanation of the following might be useful: “Experts will be asked to strongly consider the priority outcomes for weight management reporting in this round”
--	--

REVIEWER	Elizabeth Sturgiss Australian National University, Australia
REVIEW RETURNED	21-Aug-2018

GENERAL COMMENTS	This is a protocol for the development of Core Outcome measures for weight management interventions. It has been registered with COMET. This is important work and this protocol should be published. There are a few suggestions that I offer in the spirit of academic collegiality. 1. There is an existing core outcome set, with COMET, for bariatric surgery interventions. This should be stated (and cited) in the background and reasons for needing a different set of outcomes given. Why did you choose to do a new Delphi process and not adapt the existing set? 2. The restriction of the Delphi to the UK seems problematic for this international area of research. It is not clear why this was done as the Delphi is being conducted online. This should be justified. Does this mean the outcome set will only be applicable to the UK? 3. Patient involvement is minimal - there are only 10 patients involved in the first round Delphi, and none in the second round. As Patient Reported Outcomes Measures are now gaining significant traction in healthcare, this seems to be a low level of involvement from patients. This is a limitation that should be discussed. Further, the people are identified as "members of the public" -will these be people who are affected by obesity or a weight related health condition? 4. Definition of patient population - it is unclear if you are interested in obesity, or diabetes, or metabolic syndrome, or something else?
---

	This decision would greatly impact the appropriateness of chosen outcome measures. 5. As the initial Delphi only had existing studies feeding into the process, this limits any ability to extend, change, or adapt the way weight management programs are being evaluated. It would seem to be a limitation to only be assessing outcomes that are currently used, rather than thinking broadly about the things that could be done better. This is particularly apparent when there is low involvement of patients (see point 3). 6. The reference list is sparse - depending on whether you are interested in obesity/diabetes/metabolic syndrome, there are many references missing as none of these are discussed. 7. Although there is discussion in the literature on Delphi sample sizes, there are papers that offer guidance. No citations are given for the authors current stance "There is no published agreement on the optimal size of an expert group; pragmatism is required while ensuring a range of opinions is garnered." Papers do exist that assist in determining and justifying varying sample sizes for the Delphi method.
--	--

REVIEWER	Ewelina Rogozinska Queen Mary University of London, United Kingdom I am affiliated with the CoRe Outcome Set in Women and Newborn health initiative and have no other competing interests to declare.
REVIEW RETURNED	30-Aug-2018

GENERAL COMMENTS	Mackenzie et al. work submitted to BMJ Open is a well designed and written protocol of a core outcome set. My remarks are only minimal but hopefully will add to the clarity of outlined work. It is only through the reference to the NICE guidance that we learn that the core outcome set (COS) is for the lifestyle weight management programmes for overweight and obese adults. I would suggest making it more explicit in the title, abstract and the study objectives for which group of service users the COS is being developed, unless authors see it being applicable across the board (children, pregnant women), then it should also be clearly described in the protocol. Can you please provide more details on the type of study/evaluations for which the COS is being developed as it not clear to me from the current version of the protocol. Page 9 Sampling Framework - It seems that the work is conducted only within the UK settings. Can you please comment on the external validity of your outputs? Will this work will be relevant to an evaluation of lifestyle weight management programmes in developing countries, e.g. Brazil or China? On page 11 under the heading "Stage 1, Round 1 Delphi" the authors specify that the outcomes will be grouped under appropriate domains". Does the research team plan to use any form of an external framework like OMERACT filter 2.0 or will group the outcomes based on experts opinion? On pg 12 the authors write as follows: " Using the consensus on the outcome set size and importance of outcomes, an outcome set will be developed...". My question here is doing the research team have anything in place to deal with a situation of multiple outcomes
---

	of the comparable or the same importance. It is not uncommon for COS to list up to 20 - 30 outcomes. From my point of view, this makes COS highly infeasible and decreases the probability of their uptake. Finally, on pg 14 I would split the section Ethics and dissemination into two separate sections. I would like to know more about the implementation strategy once the core outcome set is developed as COS not put into action are a mere art for the sake of art.
--	--

VERSION 1 – AUTHOR RESPONSE

Reviewer 1

Reviewer Name: Jamie Hartmann-Boyce

Institution and Country: University of Oxford, UK Please state any competing interests or state 'None declared': None declared

This study is much needed and will no doubt make a valuable contribution to this field of research - I commend the authors on carrying this out. My comments are attached under separate cover (note - I am not familiar with the statistical methods for evaluating Delphi surveys and therefore am unable to comment on these, though from a non-expert point of view they seem appropriate). Most of my comments are minor and are "nice-to-have" rather than "need-to-have"; however, I do think increasing the level of involvement from patients and members of the public would markedly strengthen this work, and so have suggested this as a major revision, as I am keen that the authors consider this (and if resources don't allow further involvement, explicitly state this). Overall, though, I think this study will make a valuable contribution regardless of whether my suggested changes are made.

Response: We thank Reviewer 1 for these encouraging comments.

General (throughout)

Reviewer 1: There appears to be growing consensus that it is more appropriate to refer to these types of

programmes as "behavioural weight management programmes" as opposed to "lifestyle weight management programmes", as the latter is slightly more vague and difficult to interpret. Would suggest the authors consider changing this terminology.

Response: We thank Reviewer 1 for this suggestion and have changed the terminology throughout the manuscript.

Reviewer 1: I think the findings from this exercise will be more valuable if patients/members of the public are included in a more robust way. I have the following suggestions:

1. include public advisors as part of project team, particularly to review recruitment materials and Delphi survey text before it is sent out to ensure it is written in accessible language (experience from other studies suggests when academics write in "lay language" it can still be a bit tricky to interpret!);

Response: We fully understand the concerns of Reviewer 1 and have addressed these concerns under the 'Reviewers' Comments – General' and 'Reviewers' Comments – Public Involvement' sections above.

2. search to see if any work has been done with patient groups to establish outcomes or key questions in this area, rather than just relying on published literature and work from public health bodies (e.g. James Lind Alliance – if such work does not exist and resource permits the authors may want to consider a focus group with members of the public to identify new outcomes that aren't in the literature, as I think doing this via Delphi may be challenging);

Response: Please see the 'Reviewers' Comments – Public Involvement' section above.

3. involve patients (and primary care staff) in second stage of Delphi (their opinions on measurement seem valid and I'm not clear why you haven't included these groups);
Response: Please see the 'Reviewers' Comments – Public Involvement' section above.

4. Consider involving 20 (rather than 10) members of public in the Delphi exercise.
Response: Please see the 'Reviewers' Comments – Public Involvement' section above.

Specific

Abstract: "very few real-world programmes have published outcomes for patients" – suggest rewording – do you mean very few have published outcomes for patient consumption (e.g. an outcome report aimed at patients) or that very few have published patient-related outcomes?

Response: We thank Reviewer 1 for raising this point and have now changed this wording to 'very few have published patient-related outcomes' in order to clarify our point.

Strengths and limitations of this study – suggest removing "minor" from the phrase "some minor adaptation" as without conducting a broader exercise it is difficult to anticipate the extent of adaptation that may be needed

Response: We agree with Reviewer 1 and have now removed the word 'minor' from this sentence.

Stage 1 Delphi process – further explanation of the following might be useful: "Experts will be asked to strongly consider the priority outcomes for weight management reporting in this round"

Response: We appreciate that this might not be entirely clear and, as such, we have expanded the section in question (lines 322-326). We now explain that should we find at this point in the stage 1 Delphi that a large number of outcomes are still being rated as important, we will reinforce to participants the need to decide which outcomes should take priority and rate only these outcomes as important in the final round. This will help us to develop a core outcome set of a manageable/practical size.

Reviewer: 2

Reviewer Name: Elizabeth Sturgiss

Institution and Country: Australian National University, Australia Please state any competing interests or state 'None declared': None declared

This is a protocol for the development of Core Outcome measures for weight management interventions. It has been registered with COMET.

This is important work and this protocol should be published. There are a few suggestions that I offer in the spirit of academic collegiality.

Response: We thank Reviewer 2 for these encouraging comments.

1. There is an existing core outcome set, with COMET, for bariatric surgery interventions. This should be stated (and cited) in the background and reasons for needing a different set of outcomes given. Why did you choose to do a new Delphi process and not adapt the existing set?

Response: We thank Reviewer 2 for this suggestion and agree that the BARIACT project should be mentioned. We have now referred to this study in the Introduction section of our manuscript (lines 100-102).

With regard to why we chose a new Delphi process, the COS developed during the BARIACT project is recommended to be used as a minimum in all trials of bariatric and metabolic surgery and focusses

on outcomes including perioperative outcomes and post-operative complications which are not relevant for behavioural weight management programmes.

2. The restriction of the Delphi to the UK seems problematic for this international area of research. It is not clear why this was done as the Delphi is being conducted online. This should be justified. Does this mean the outcome set will only be applicable to the UK?

Response: Please see 'Reviewers' Comments – UK Setting' section above.

3. Patient involvement is minimal - there are only 10 patients involved in the first round Delphi, and none in the second round. As Patient Reported Outcomes Measures are now gaining significant traction in healthcare, this seems to be a low level of involvement from patients. This is a limitation that should be discussed. Further, the people are identified as "members of the public" -will these be people who are affected by obesity or a weight related health condition?

Response: Please see 'Reviewers' Comments – Public Involvement' section above.

With regard to the members of the public participating in our study, as stated in the first paragraph of the 'Patient and Public Involvement' section (lines 162-164), these will be people with experience of NHS, local authority or commercial adult behavioural weight management programmes in the UK. Our sampling framework is also outlined further on in the manuscript (lines 222-229).

4. Definition of patient population - it is unclear if you are interested in obesity, or diabetes, or metabolic syndrome, or something else? This decision would greatly impact the appropriateness of chosen outcome measures.

Response: As stated in our Introduction section, we are not interested in any of these morbidities specifically but rather in developing a core outcome set in order to standardise reporting of results from behavioural weight management programmes which have the primary purpose of weight loss regardless of underlying comorbidities.

The core outcome set will allow behavioural weight management programmes to be compared more effectively which, in turn, will reveal which programmes work best for which members of the population, ensuring the improvement of UK weight management programmes in general.

5. As the initial Delphi only had existing studies feeding into the process, this limits any ability to extend, change, or adapt the way weight management programs are being evaluated. It would seem to be a limitation to only be assessing outcomes that are currently used, rather than thinking broadly about the things that could be done better. This is particularly apparent when there is low involvement of patients (see point 3).

Response: Please see 'Reviewers' Comments – Public Involvement' section above.

6. The reference list is sparse - depending on whether you are interested in obesity/diabetes/metabolic syndrome, there are many references missing as none of these are discussed.

Response: As previously mentioned, we are not specifically interested in any of these morbidities. Our reference list reflects our interest in previous attempts to standardise reporting from UK behavioural weight management programmes and in the outcomes previously reported from such programmes in real world clinical practice as well as within research studies.

7. Although there is discussion in the literature on Delphi sample sizes, there are papers that offer guidance. No citations are given for the authors current stance "There is no published agreement on the optimal size of an expert group; pragmatism is required while ensuring a range of opinions is

garnered." Papers do exist that assist in determining and justifying varying sample sizes for the Delphi method.

Response: We thank Reviewer 2 for this comment and have sought advice from our colleagues at Clinivo who believe that our current stance is sensible and similar to that of the BARIACT group (Coulman et al. 2016) who state that, 'There is no specific guidance for determining the sample size required for Delphi surveys. For this study, a sample of 100 professionals and 100 patients was anticipated to provide a representative sample including a broad set of views.'

We believe it is important to note that when we carry out a Delphi, we are not 'sampling' from a population and then trying to measure something about that sample to estimate a population parameter through statistical inference i.e. we are not trying to (and not able to) externally generalise to a wider population. Thus, there is no sample size calculation as there is no hypothesis to test and traditional confidence intervals are therefore not appropriate to accompany estimation either.

Increasingly, authors seem to be referring to Delphi 'surveys'. However, Delphi studies are not really surveys – surveys are empirical studies. In Delphi, questionnaires are used but this is really all that the two approaches have in common. Questionnaires are used so that the experts do not need to meet (and so can feel free to give their unfettered and uncensored opinions).

While there may be other papers in the literature considering approaches to informing expert panel size, there cannot be any solid empirical formula to calculate a 'sample' size or it cannot be a Delphi/modified Delphi as it is adopting an empirical framework for something that is inherently unempirical.

Reviewer: 3

Reviewer Name: Ewelina Rogozinska

Institution and Country: Queen Mary University of London, United Kingdom Please state any competing interests or state 'None declared': I am affiliated with the CoRe Outcome Set in Women and Newborn health initiative and have no other competing interests to declare.

Mackenzie et al. work submitted to BMJ Open is a well designed and written protocol of a core outcome set. My remarks are only minimal but hopefully will add to the clarity of outlined work.

Response: We thank Reviewer 3 for these encouraging comments.

It is only through the reference to the NICE guidance that we learn that the core outcome set (COS) is for the lifestyle weight management programmes for overweight and obese adults. I would suggest making it more explicit in the title, abstract and the study objectives for which group of service users the COS is being developed, unless authors see it being applicable across the board (children, pregnant women), then it should also be clearly described in the protocol.

Response: We thank Reviewer 3 for bringing this oversight to our attention. We agree that this should be made more explicit. We have amended our title to include the relevant information and it now reads, 'Development of a core outcome set for behavioural weight management programmes for adults with overweight and obesity: protocol for obtaining expert consensus using Delphi methodology'. We have also made reference to the specific group of service users for which the COS is being developed (adults with overweight and obesity) in the Abstract, Introduction and Study Objectives.

Can you please provide more details on the type of study/evaluations for which the COS is being developed as it not clear to me from the current version of the protocol.

Response: As stated in the Introduction (lines 102-105), the COS is being developed for behavioural weight management programmes in real world clinical practice as well as within research studies. These programmes tend to include calorie restriction, increased physical activity, and behavioural

interventions and may be commercial or non-commercial, such as those delivered in primary care settings (for example, in GP practices).

Page 9 Sampling Framework - It seems that the work is conducted only within the UK settings. Can you please comment on the external validity of your outputs? Will this work will be relevant to an evaluation of lifestyle weight management programmes in developing countries, e.g. Brazil or China?
Response: Please see 'Reviewers' Comments – UK Setting' section above.

On page 11 under the heading "Stage 1, Round 1 Delphi" the authors specify that the outcomes will be grouped under appropriate domains". Does the research team plan to use any form of an external framework like OMERACT filter 2.0 or will group the outcomes based on experts opinion?

Response: We did not use an external framework here but rather grouped outcomes under general headings/domains which were broadly based on the PHE SEF and followed the weight management chronological pathway. This seemed the most natural way to do this. The domains included, 'Demographics', 'Physical Measurements', 'Physical Activity', 'Diet', 'Comorbidities', 'Lifestyle Behaviours' and 'Psychological Factors', and, as outlined, all domain headings were fully explained in lay language. Information to this effect has now been added to the manuscript (lines 297-299).

On pg 12 the authors write as follows: " Using the consensus on the outcome set size and importance of outcomes, an outcome set will be developed...". My question here is doing the research team have anything in place to deal with a situation of multiple outcomes of the comparable or the same importance. It is not uncommon for COS to list up to 20 - 30 outcomes. From my point of view, this makes COS highly infeasible and decreases the probability of their uptake.

Response: We thank Reviewer 3 for this question and have now amended the paragraph preceding the one she highlights (lines 319-325) to read:

'Stage 1, Round 3 Delphi

In round 3, all experts will be asked to rate outcomes for the final time. They will be shown their previous rating, the median expert group rating and any free text comments in the hope of ratings reaching a consensus. Should it be the case that a large number of outcomes are being rated as important at this stage, the need to decide which outcomes should take priority for weight management reporting will be reinforced to experts and they will be asked to rate only these priority outcomes as important. This will ensure development of a core outcome set of a manageable/practical size.'

Finally, on pg 14 I would split the section Ethics and dissemination into two separate sections. I would like to know more about the implementation strategy once the core outcome set is developed as COS not put into action are a mere art for the sake of art.

Response: We thank Reviewer 3 for this suggestion and have now split this section.

As stated in this section, we will share our findings with PHE and Health Scotland. We will be in full discussion with both bodies so that our work can inform their evaluation plans for weight management services. We believe our core outcome and definition/instrument sets will be suitable for international use with some adaptation. We have now added this information to the 'Dissemination' section. We have also outlined our plans to publish our findings in a peer-reviewed journal and to present our work at research conferences and professional societies.

VERSION 2 – REVIEW

REVIEWER	Jamie Hartmann-Boyce University of Oxford
REVIEW RETURNED	02-Nov-2018

GENERAL COMMENTS	The authors have done a thorough job addressing comments and I am happy with the paper as is, but suggest the authors might want to consider rephrasing the below newly added line (177): "With such a level of knowledge and expertise required, members of the public and primary care staff will not be involved in this stage of the Delphi process" as it risks possibly offending members of the public or primary care staff. Suggest it might be more politic to rephrase to something along the lines of "As this will require in-depth knowledge of academic literature and reporting tools in this area, we are restricting this phase of the process to academics, policy makers, commissioners, and weight management staff." In addition, line 388 needs "is" to be added ("it is our belief").
---

REVIEWER	Elizabeth Sturgiss The Australian National University
REVIEW RETURNED	29-Oct-2018

GENERAL COMMENTS	Thank you for your responses to the reviewer comments. There are two issues that I believe need to be addressed prior to publication. I refer to my original comment: "As the initial Delphi only had existing studies feeding into the process, this limits any ability to extend, change, or adapt the way weight management programs are being evaluated. It would seem to be a limitation to only be assessing outcomes that are currently used, rather than thinking broadly about the things that could be done better. This is particularly apparent when there is low involvement of patients (see point 3). Response: Please see 'Reviewers' Comments – Public Involvement' section above." I don't believe the authors have addressed my concern and it is not discussed in the public involvement section. As you are only considering existing outcomes from existing studies there is no option for your outcome set to improve the status quo. I believe this could be addressed by allowing for the Delphi process to consider outcomes that are not currently used but that you think have been overlooked. If it is not possible for this to happen as you have already commenced, I believe it should be in the limitations of your study. Secondly, I do not agree with logic presented about not having international participants. It is completely fine to have a UK specific development and as you have mentioned, there are context-specific factors. However, it cannot then be said that the outcome set is likely to be applicable internationally - it can only be one or the other. It should be added to your limitations that this is a UK specific process and international translation may be required. I think it is an over-reach to say "However, it our belief that these differences are subtle and that, particularly for trials, our core outcome and definition/instrument sets could be used internationally with some adaptation." I think this statement should be removed as it does not follow the logic of the preceding argument. I think it is fine to leave it as a limitation and say that internationally it may require translation. Also my suggestions for the Strengths and Limitations: It is a limitation that this study is wholly based in the United Kingdom (UK) as the results may need some adaptation to be
---

	suited to real-world programmes set within other healthcare systems. - I AGREE WITH THIS STATEMENT However, we will use the internationally recognised Delphi method to garner opinions from a wide range of individuals with expertise in behavioural weight management. - I DO NOT THINK YOU SHOULD USE THE WORD "INTERNATIONALLY" AS IT MISLEADS THE READER (sorry the capitalisation, it is the only way to mark my feedback) This is very important work and I look forward to seeing it published.
--	---

VERSION 2 – AUTHOR RESPONSE

Reviewer: 1

Reviewer Name: Jamie Hartmann-Boyce

Institution: University of Oxford

Please state any competing interests or state 'None declared': None declared

The authors have done a thorough job addressing comments and I am happy with the paper as is, but suggest the authors might want to consider rephrasing the below newly added line (177): "With such a level of knowledge and expertise required, members of the public and primary care staff will not be involved in this stage of the Delphi process" as it risks possibly offending members of the public or primary care staff. Suggest it might be more politic to rephrase to something along the lines of "As this will require in-depth knowledge of academic literature and reporting tools in this area, we are restricting this phase of the process to academics, policy makers, commissioners, and weight management staff." In addition, line 388 needs "is" to be added ("it is our belief").

Response: We thank Reviewer 1 for this comment and have now changed the newly added line (lines 185-187) to read,

'As in-depth knowledge of academic literature and reporting tools is required, this stage of the Delphi process will be restricted to academics/policy makers/commissioners and weight management staff.'

Line 388 (now line 402) has been changed to address the 2nd reviewer's comments but we thank Reviewer 1 for highlighting this oversight.

Reviewer: 2

Reviewer Name: Elizabeth Sturgiss

Institution: The Australian National University Please state any competing interests or state 'None declared': None declared

Thank you for your responses to the reviewer comments. There are two issues that I believe need to be addressed prior to publication.

I refer to my original comment:

"As the initial Delphi only had existing studies feeding into the process, this limits any ability to extend, change, or adapt the way weight management programs are being evaluated. It would seem to be a limitation to only be assessing outcomes that are currently used, rather than thinking broadly about the things that could be done better. This is particularly apparent when there is low involvement of patients (see point 3).

Response: Please see 'Reviewers' Comments – Public Involvement' section above."

I don't believe the authors have addressed my concern and it is not discussed in the public involvement section.

As you are only considering existing outcomes from existing studies there is no option for your outcome set to improve the status quo. I believe this could be addressed by allowing for the Delphi process to consider outcomes that are not currently used but that you think have been overlooked. If it is not possible for this to happen as you have already commenced, I believe it should be in the limitations of your study.

Response: We thank Reviewer 2 for this comment. We appreciate that we have only included existing outcomes in our original stage 1 (outcome selection) questionnaire. However, many of the studies included in our systematic review have patient and public involvement. In addition, while we were not able to review all existing qualitative research, we carried out qualitative work during the core outcome set development as part of the Delphi process. We draw the reviewer's attention to the fact that during each round of the stage 1 Delphi, participants were given the option to suggest additional outcomes which, if considered appropriate, were carried forward to the next round of the process. This is stated within the manuscript (line 305, line 310, line 320). As such, our core outcome set may very well include outcomes not currently being used.

We have now added the following to the limitations section (lines 52-54),

'Review of all existing qualitative research studies will not be undertaken when generating the initial list of outcomes. However, qualitative work will be performed during core outcome set development as part of the Delphi process.'

With regard to our outcome set failing to improve the status quo, there is currently no core outcome set in place for behavioural weight management programmes (BWMPs). Therefore, it seems likely that our core outcome set cannot fail to improve the status quo by reducing the heterogeneity of reporting.

Reviewer: Secondly, I do not agree with logic presented about not having international participants. It is completely fine to have a UK specific development and as you have mentioned, there are context-specific factors. However, it cannot then be said that the outcome set is likely to be applicable internationally - it can only be one or the other. It should be added to your limitations that this is a UK specific process and international translation may be required. I think it is an over-reach to say "However, it our belief that these differences are subtle and that, particularly for trials, our core outcome and definition/instrument sets could be used internationally with some adaptation." I think this statement should be removed as it does not follow the logic of the preceding argument. I think it is fine to leave it as a limitation and say that internationally it may require translation.

Response: We thank Reviewer 2 for this comment and have now changed this (lines 402-404) to read,

'Therefore, if used in an international context for trials or real world services, our core outcome and definition/instrument set may require further adaptation. '

Reviewer: Also my suggestions for the Strengths and Limitations:

It is a limitation that this study is wholly based in the United Kingdom (UK) as the results may need some adaptation to be suited to real-world programmes set within other healthcare systems. - I AGREE WITH THIS STATEMENT However, we will use the internationally recognised Delphi method to garner opinions from a wide range of individuals with expertise in behavioural weight management. - I DO NOT THINK YOU SHOULD USE THE WORD "INTERNATIONALLY" AS IT MISLEADS THE READER (sorry the capitalisation, it is the only way to mark my feedback)

Response: We thank Reviewer 2 for this comment and agree that this could be misleading. We wanted to make the point that the Delphi method is the internationally recognised method for core outcome set development. We have now changed this section (lines 49-51) to read,

'The recognised method for core outcome set development, the Delphi method, will be used to garner opinions from a wide range of individuals with expertise in behavioural weight management.'

This is very important work and I look forward to seeing it published.